# How Do Problem-Solving Demands Influence Employees' Thriving at Work: An Explanation Based on Cognitive Appraisal

Lulu Ma [1,2,3], Hongyu Ma [1,2,*], Xiangping Zhan [1,2] and Yue Wang [1,2]

1 School of Psychology, Central China Normal University, Wuhan 430079, China; xiangjiaoshu247@163.com (L.M.)
2 Key Laboratory of Adolescent Cyberpsychology and Behavior (CCNU), Ministry of Education, Wuhan 430079, China
3 Institute of Marxism, Shijiazhuang Posts and Telecommunications Technical College, Shijiazhuang 050020, China
* Correspondence: mahy@mail.ccnu.edu.cn

**Abstract:** In globalized markets, it is important for companies to cultivate a thriving workforce that is motivated to grow and develop. Based on the transactional theory of stress, we discussed how the way people appraise their problem-solving demands, either as a challenge or a hindrance, impacts employees' thriving at work. Data were collected from employees of a state-owned enterprise in China at two separate points with a 4-week interval. The results showed that problem-solving demands have a positive impact on employees' thriving at work through challenge appraisal and a negative impact on employees' thriving at work through hindrance appraisal. Additionally, we observed a moderated mediation effect in which organizational identity strengthened the positive effects of problem-solving demands on challenge appraisal, which in turn promoted employees' thriving at work. The findings highlight the role of cognitive appraisal in interpreting employees' responses to work stress.

**Keywords:** problem-solving demands; thriving at work; challenge appraisal; hindrance appraisal; organizational identity





## 1. Introduction

In globalized markets, in order to adapt quickly to changing and uncertain environments, companies must maintain competitiveness. It is becoming increasingly important for companies to cultivate a thriving workforce that is motivated to grow and develop [1]. Thriving at work was described as "a psychological state in which individuals experience both a sense of vitality and learning at work" [2]. Research finds that when an organization has thriving employees and work teams, it becomes more agile and resilient in the face of challenges and crises, and more adaptable to uncertain external environments [3]. Thriving at work is a positive work state and serves as a crucial psychological motivation for employee growth and development, which can improve employees' performance and well-being and promote organizational performance and sustainable development [1,4,5].

How to facilitate employees' thriving in an increasingly complex work environment is a focal point of management practice. The complexity of the work environment poses numerous new challenges and issues, with problem-solving demands (PSDs) emerging as a common source of stress [6,7]. Despite extensive research on stress management, little attention has been paid to knowledge-related demands, such as PSDs [8]. Although individual and relational available resources are positively related to thriving at work [5], the impact of job stressors on thriving has not received sufficient attention. Spreitzer et al. pointed out that "the contextual factors of thriving are not merely the opposite factors that exacerbate stress" and "thriving is not cultivated simply by decreasing stressors" [2]. This assumption implicitly suggests that job stressors have negligible or negative effects on employees' thriving, but they never explicitly ruled out that some job stressors could be potential facilitators of thriving at work [9]. For example, Prem et al. found that both time

pressure and learning demands could directly and indirectly (through challenge appraisal) positively predict learning (rather than vitality), and only learning demands negatively predict vitality through hindrance appraisal [9]. Prem et al. investigated how demands link to the different dimensions of thriving through three daily surveys and found limited evidence of demands on vitality. It may be that the time interval is too short or ignores the role of the moderation variable. Thus, the role of job stressors, especially knowledge-related challenge stressors (such as PSDs), in relation to employees' thriving at work needs further exploration.

As for PSDs, the research reveals the double-sword effect of it. Some studies have found that PSDs were positively related to employees' creativity and work engagement [10–12], while other studies have found that PSDs had no significant relationship with work engagement and job satisfaction but can significantly predict psychological strain and is negatively related to employee well-being [8,13]. Espedido and Searle built two paths to clarify the effects of PSDs based on the transactional theory of stress. Studies revealed that through challenge appraisal, PSDs were positively predicted proactive behavior (i.e., innovation, voice, problem prevention), and through threat appraisal, PSDs were positively predicted undermining behavior; at the same time, psychological safety climate could strengthen the relationship of PSDs and challenge appraisal [7]. Espedido's work enlightens us to clarify the relationship between PSDs and thriving through the cognitive appraisal mechanism. Espedido et al. focus on the behavior that PSDs bring, while we pay more attention to the psychological states that PSDs cause.

As mentioned above, the transactional theory of stress [14] is a useful theoretical framework to clarify the different effects of PSDs. The transactional theory of stress points out that individuals' behavioral responses to stressors are explained by the way people appraise (interpret) them [14]. When encountering a stressor, individuals first evaluate the meaning and significance of a situation. If the stressful encounter touches something important to the individual, he/she will make appraisals to frame its meaning in relation to himself/herself (i.e., challenging or hindering) [14]. Therefore, the current research focuses on the role of cognitive appraisal in the relationship between PSDs and employees' thriving at work.

Moreover, the transactional theory of stress highlights the influence of both individual and situational factors on an individual's appraisal of stress [14,15]. Organizational identity can be regarded as an effective situational resource that can alter employees' appraisal of PSDs. Organizational identity is the sense of unity that employees share weal and woe with the organization. With a stronger sense of organizational identity, employees are more likely to uphold the interests of the organization, which is also a key driver of employees' organizational behavior [16,17]. How will organizational identity, as an organization-related employee self-perception, affect employees' appraisal of job demands (such as PSDs), and subsequently influence their thriving at work? The present study further examines how organizational identity moderates the relationship between PSDs and thriving at work through cognitive appraisal.

### 1.1. Cognitive Appraisal as the Mechanisms between PSDs and Thriving at Work

Problem-solving demands refer to the extent to which a job requires employees to actively utilize their knowledge and skills to "diagnose and solve problems" at work [18] (p. 208), and it reflects the degree to which the job requires new ideas and solutions to non-routine and ambiguous problems, thereby challenging employees to develop new solutions to problems, stretching their knowledge and skill bases [10]. PSDs is a specific aspect of job complexity, but it captures the extent to which the job requires the individual to develop new and useful solutions to problems. PSDs also contain some elements of innovation, but it differs from an employee's motivation to develop creative problem solutions, because PSDs pertain to the extent to which the job demands the individual to develop skills and new solutions to problems. Whereas the effects of stressors traditionally considered as

sources of distress (e.g., time pressure, workload) on stress appraisals have been explored, appraisals of knowledge-related demands such as PSDs are less understood [8].

Because PSDs are related to task complexity and contains some elements of innovation [19,20], in the challenge–hindrance framework, PSDs are classified as a challenge stressor. Studies have found that PSDs can indeed improve employees' creativity [10,11]. However, the relationship between PSDs and other organizational performance outcomes and well-being is unclear. Based on the job demand–resources model, Huo and Boxall found that PSDs, as a challenging demand, can promote work engagement, but has no significant relationship with exhaustion [12]. Based on the same model, another study found that job demands (including PSDs) negatively predicted exhaustion but had no significant relationship with work engagement [13]. Based on the conservation of resource theory, Schmitt et al. found that PSDs had no significant relationship with job satisfaction but would cause fatigue. The above research results are contradictory and accidental [8]. Espedido et al. explored the relationship between PSDs and proactive behaviors based on the transactional theory of stress and found that appraisals of challenge mediated the relationship between PSDs and favorable forms of proactivity (innovation and voice behavior), whereas appraisals of threat mediated the relationship with unfavorable forms of proactivity (undermining) [6,7].

The stressors categorized as "challenges", such as time pressure and workloads, have been appraised as both challenging and a hindrance [21,22], while it is still unclear whether the challenging stressor PSDs are generally perceived as challenging. A meta-analysis indicates that while both types of job stressors have adverse effects, challenge stressors (as opposed to hindrance stressors) also have favorable effects on motivation and performance [23–25]. Conceptual and empirical research suggests that PSDs may be appraised as both challenging and threatening [14,26]. Appraisal is essential to explain the relationship between PSDs and thriving at work, because interpretations of PSDs are "likely to vary, as they might be experienced as challenging and motivating, but also as adverse and hindering" [26].

PSDs can be challenging for employees because they require people to stretch their knowledge and skill base to diagnose non-routine problems at work [18]. If employees see opportunities for learning and growth in problem-solving, they will strive to learn new skills and constantly try to solve problems. In the process of problem-solving, employees continue to learn and maintain vitality to overcome difficulties and solve problems, thus improving thriving at work. At the same time, PSDs may also tax mental resources and have been linked to personal losses, such as fatigue and lower levels of available cognitive resources [8,27]. PSDs may present interpersonal risks because employees could face embarrassment, negative evaluation or rejection from colleagues if they do not solve problems effectively [7]. As a result, employees who perceive PSDs as a hindrance may refrain from engaging in learning and problem-solving activities, leading to demotivation, which hinders their ability to thrive at work.

**H1a.** *Challenge appraisal will mediate the positive relationship between PSDs and thriving at work.*

**H1b.** *Hindrance appraisal will mediate the negative relationship between PSDs and thriving at work.*

### 1.2. The Moderating Role of Organizational Identity

Espedido et al. emphasized that the growing interdependence among contemporary employees and the significant influence of environmental factors and team climate in shaping employee attitudes and perceptions should not be disregarded [6,7]. Their research revealed that psychological safety climate and team problem prevention can moderate the relationship between PSDs and cognitive appraisal. In the face of new work tasks, individuals' problem-solving approaches at work are largely shaped by their organizational identities [28]. Organizational identity refers to how employees "see themselves and the organization as one and empathize with the success and failure of the organization"; it means that when employees' organizational identity is higher, they have stronger motivation to

seek benefits for the organization [16]. They will internalize their group's norms, values and goals and strive towards maintaining a positive evaluation of the group [28]. Due to the strong link with the organization, employees tend to perceive PSDs as an indispensable requirement for personal and organizational development. They are more likely to see the learning and growth brought by problem-solving, so they tend to make challenge appraisals. On the contrary, individuals with low organizational identity are more likely to see various risk factors encountered in the process of problem-solving. They tend to adhere to routine and avoid making additional efforts to solve problems, so they are more likely to make hindrance appraisals.

**H2a.** *Organizational identification moderates the relationship between PSDs and challenge appraisal. The higher the organizational identification, the relationship between PSDs and hindrance appraisal will be stronger.*

**H2b.** *Organizational identification moderates the relationship between PSDs and hindrance appraisal. The higher the organizational identification, the relationship between PSDs and hindrance appraisal will be weaker.*

One study found that psychological safety climate strengthened the positive effects of within-person problem-solving demands on challenge appraisal, which in turn promoted proactive innovation [7]. With a strong sense of organizational identity, employees are more likely to focus on problem-solving and exhibit greater resilience, thus bringing learning and growth. On the contrary, employees with low organizational identity will view PSDs as a source of additional work pressure, leading them to avoid investing time and effort in problem-solving and thereby hindering their ability to thrive at work. A conceptual model is shown in Figure 1.

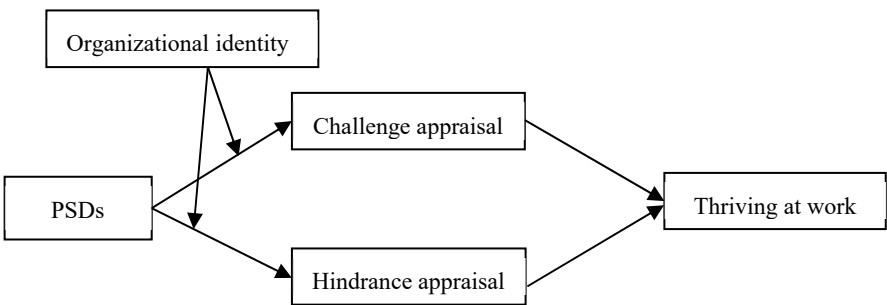

**Figure 1.** Conceptual model.

**H3a.** *Organizational identity will moderate the positive effects of problem-solving demands on thriving via challenge appraisal, such that the positive relationship between problem-solving demands and challenge appraisal will be stronger at higher levels of organizational identity.*

**H3b.** *Organizational identity will moderate the negative effects of problem-solving demands on thriving via hindrance appraisal, such that the positive relationship between problem-solving demands and challenge appraisal will be weaker at higher levels of organizational identity.*

## 2. Materials and Methods

### 2.1. Sample and Procedure

Participants were recruited through the personal networks of the first author and an undergraduate student. All participation was voluntary. The research purpose and procedures were briefly explained to potential participants, who were then invited to join our QQ group. The participants were appropriately compensated as a token of our gratitude for their valuable contribution. The data were collected through a two-stage online questionnaire with a four-week interval. Demographic variables, PSDs, cognitive appraisal and organizational identification were collected in the first questionnaire, and 383 valid

data points were collected. Four weeks later, we collected thriving in the second questionnaire, and 304 valid data points were collected. Finally, 211 valid questionnaires were successfully matched, and the matching rate was 54.13%. In the final sample (67.8% male and 32.2% female), the mean age was 25.75 years (SD = 4.47). The mean job tenure was 19.49 months (SD = 19.90) and the mean working time was 51.24 h per week (SD = 12.74). As expected, the education levels of our sample were relatively high due to the recruitment from state-owned enterprises: 75 junior college students (35.5%), 130 bachelor students (61.6%) and 6 graduate students (2.8%).

## 2.2. Measures

Following translation/back-translation procedures, all items were translated into Chinese. The items were rated on a five-point Likert scale (1 = not at all/strongly disagree; 5 = very frequently/strongly agree).

Problem-solving demands. Problem-solving demands were measured using Wall et al.'s (1996) [29] five-item scale. A sample item is "To what extent have you been required to solve problems which have no obvious correct answer?" (Cronbach's $\alpha$ = 0.64).

Challenge and hindrance appraisals. Challenge and hindrance appraisals were measured by using the two four-item scales developed by Searle and Auton (2015) [21]. Respondents were asked to think about the problem-solving demands and how they are likely to affect them. A sample item of challenge appraisal is "It will help me to learn a lot" (Cronbach's $\alpha$ = 0.62). A sample item of hindrance appraisal is "It will hinder any achievements I might have" (Cronbach's $\alpha$ = 0.90).

Organizational identification. Organizational identification was measured with five items from a scale developed by Mael and Ashforth (1992) [30]. These items assess the extent to which employees identify with their organizations. A sample item is "The organization's successes are my successes" (Cronbach's $\alpha$ = 0.83).

Thriving at work. Thriving at work was measured by using the eight positively worded items from Porath and colleagues' (2012) [4] scale. A sample item of learning is "Today, I have developed as a person" and a sample item of vitality is "Right now, I feel alive and vital" (Cronbach's $\alpha$ = 0.89).

## 2.3. Data Analysis

Firstly, confirmatory factor analysis (CFA, using MPlus version 8.3) was used to test the proposed factor structures of all study variables. The results of confirmatory factor analysis showed that the five-factor model fitted the data best ($\chi^2$ = 571.729, df = 289, RMSEA = 0.069, SRMR = 0.084, CFI = 0.884, TLI = 0.870), and the fitting results were better than those of other competing models. All items were modeled onto their respective latent factors, with the analysis partitioning variance at the item level. Harman single factor method was used to test the common method bias, and it was found that under the condition of unrotated factor, the maximum factor accounted for 31.07% of the total variation, which was less than 40% of the criterion, indicating that there were no serious common method biases in this study. Secondly, SPSS 23.0 was used for descriptive statistical analysis, and, finally, SPSS PROCESS v4.0 was used for hypothesis testing.

## 3. Results

### 3.1. Correlations

As shown in Table 1, PSDs were significantly positively associated with both challenge appraisal ($r$ = 0.177, $p$ < 0.1) and hindrance appraisal ($r$ = 0.312, $p$ < 0.01). Challenge appraisal was significantly positively associated with thriving at work ($r$ = 0.598, $p$ < 0.1), and hindrance appraisal was significantly negatively associated with thriving at work ($r$ = −0.237, $p$ < 0.01). PSDs were significantly positively associated with thriving at work ($r$ = 0.226, $p$ < 0.01).

**Table 1.** Descriptive statistics and correlations among study variables.

| | M | SD | 1 | 2 | 3 | 4 | 5 | 6 | 7 | 8 | 9 |
|---|---|---|---|---|---|---|---|---|---|---|---|
| 1. Age | 25.75 | 4.466 | | | | | | | | | |
| 2. Gender | - | - | −0.2173 ** | | | | | | | | |
| 3. Education | - | - | 0.2593 ** | −0.284 ** | | | | | | | |
| 4. Job tenure (months) | 19.485 | 19.900 | 0.550 ** | −0.052 | 0.172 * | | | | | | |
| 5. Working time (per week) | 51.24 | 12.735 | −0.041 | 0.013 | −0.115 | −0.030 | | | | | |
| 6. Problem-solving demands | 3.642 | 0.637 | 0.075 | 0.019 | 0.049 | 0.164 * | 0.150 * | | | | |
| 7. Challenge appraisal | 4.307 | 0.452 | 0.107 | −0.115 | 0.068 | 0.248 ** | −0.081 | 0.177 ** | | | |
| 8. Hindrance appraisal | 2.549 | 1.210 | −0.080 | 0.004 | −0.132 | 0.096 | 0.282 ** | 0.312 ** | −0.270 ** | | |
| 9. Organizational identification | 4.226 | 0.631 | 0.212 ** | −0.260 ** | 0.292 ** | 0.297 ** | −0.062 | 0.260 ** | 0.546 ** | −0.180 ** | |
| 10. Thriving at work | 4.190 | 0.599 | 0.218 ** | −0.164 * | 0.369 ** | 0.308 ** | −0.095 | 0.226 ** | 0.598 ** | −0.237 ** | 0.743 ** |

*Note: N = 211, * p < 0.05, ** p < 0.01.*

### 3.2. Mediated Effects of Cognitive Appraisal

Model 4 of PROCESS was used to examine the mediating role of cognitive appraisal between PSDs and thriving at work. All predictive variables were standardized. Controlling gender, age, education level, organizational tenure and working hours, PSDs were positively related to challenge appraisal ($\beta = 0.112$, $p < 0.05$), and challenge appraisal was positively related to thriving at work ($\beta = 0.641$, $p < 0.001$). Further, the results showed that the indirect effect of PSDs on thriving at work through the challenge appraisal of PSDs were significant ($\beta = 0.072$, 95%CI = [0.014, 0.138], excluding zero), and Hypothesis 1a was supported. PSDs were positively related to hindrance appraisal ($\beta = 0.514$, $p < 0.001$), and hindrance appraisal was negatively related to thriving at work ($\beta = -0.062$, $p < 0.05$). Further, the results showed that the indirect effect of PSDs on thriving at work through the hindrance appraisal of PSDs were significant ($\beta = -0.034$, 95%CI = [−0.078, −0.003], excluding zero). Hypothesis 1b was supported.

### 3.3. Moderated Mediation Effects of Organizational Identification

Model 7 of PROCESS was used to examine the moderating effect of organizational identification. Hypothesis 2a stated that the positive relationship between PSDs and challenge appraisal becomes stronger when organizational identification increases. The results showed that the interaction of PSDs and organizational identification was significant for challenge appraisal ($\beta = 0.169$, $p < 0.05$); thus, Hypothesis 2a was supported. A simple slope analysis (Figure 2) shows that for employees with higher organizational identification, PSDs have a significant positive effect on challenge appraisal ($\beta = 0.125$, t = 1.997, $p < 0.05$, 95%CI = [0.016, 0.248], excluding zero), but for employees with low organizational identity, the impact of PSDs on challenge appraisal was not significant ($\beta = -0.088$, t = −1.306, $p = 0.193$, 95%CI = [−0.221, 0.045], including zero). In summary, H2a was supported, while H2b was not supported.

We further tested whether the indirect relationships between PSDs and thriving at work via appraisals (challenge, H3a; hindrance, H3b) were moderated by organizational identity. As shown in Table 2, the indirect relationship between PSDs and thriving at work through challenge appraisal was significant in the high organizational identity group ($\beta = 0.080$, 95%CI = [0.009, 0.147], excluding zero). This indirect relationship was insignificant in the low organizational identity group ($\beta = -0.057$, 95%CI = [−0.138, 0.030], including zero). The difference between the high and low organizational identity groups was significant ($\beta = 0.108$, 95%CI = [0.015, 0.190], excluding zero). Thus, H3a was supported. In addition, we found that the indirect relationship between PSDs and thriving at work via hindrance appraisal was not significant between the high organizational identity group and the low organizational identity group ($\beta = 0.019$, 95%CI = [−0.006, 0.051], including zero). Thus, H3b was not supported.

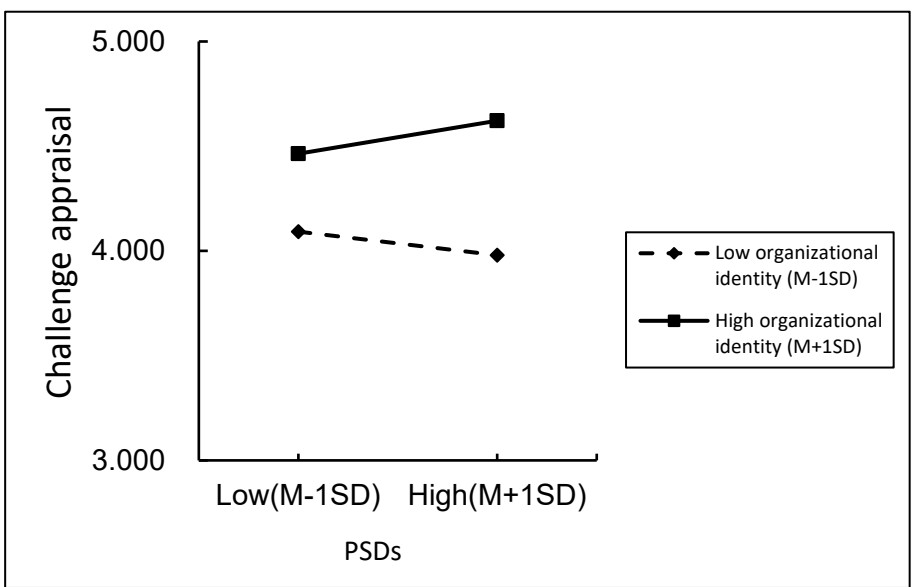

**Figure 2.** Interaction of PSDs and organizational identity on challenge appraisal.

**Table 2.** Results of indirect relationship and conditional indirect relationships.

| Path | Grouping | $\beta$ | SE | 95% CI |
|---|---|---|---|---|
| PSDs→Challenge appraisal→Thriving at work | High organizational identity | 0.080 | 0.035 | [0.009, 0.147] |
| | Low organizational identity | −0.057 | 0.042 | [−0.138, 0.030] |
| | Difference | 0.108 | 0.044 | [0.015, 0.190] |
| PSDs→Hindrance appraisal→Thriving at work | High organizational identity | −0.029 | 0.021 | [−0.080, −0.001] |
| | Low organizational identity | 0.053 | 0.028 | [−0.112, −0.004] |
| | Difference | 0.019 | 0.015 | [−0.006, 0.051] |

*Note: N* = 211.

## 4. Discussion

The main contributions of this study are as follows: Firstly, a central component of Spreitzer et al.'s socially embedded model of thriving is that "thriving can occur with or without adversity" [2]. Consistent with this assertion, the transactional theory of stress holds that the relationship between job demands and job outcomes is determined by cognitive appraisal. We directly explore the cognitive appraisal mechanisms between PSDs and thriving at work to enrich the theory of thriving and answer calls to investigate the antecedent variables of thriving at work [31]. Secondly, the previous studies about PSDs and organizational outcomes were mostly under the background of lean production [10,12,32]. PSDs require employees to engage in more active cognitive processing to diagnose and solve problems, which serves as a motivator for employees' job satisfaction [33]. However, with the development, PSDs have become a common job demand, but researchers have paid little attention to it [6,7]. We explore the mechanism between PSDs and employees' thriving at work in the context of a Chinese state-owned enterprise. A third contribution of our research is the identification of organizational identity as a boundary condition that can moderate the effects of PSDs on employee appraisals and thriving. According to the socially embedded model of thriving, Kleine et al. expected to find "a zero relationship between negative events and thriving" in their meta-analysis of the literature on thriving [5]. Instead, they found perceived stress to be negatively related to thriving. Thus, our study has the potential to inform Spreitzer et al.'s model by exploring a contextual contingency through which such null effects might be observed.

Guided by self-determination theory, Spreitzer et al. developed a socially embedded model of thriving (SEMT) that explains how individuals thrive in environments that enable them to behave agentically at work [2]. A core assumption of the SEMT is that

most employees do not work in isolation but, instead, are socially embedded in proximal work contexts (e.g., teams and work units). Work contexts that reflect trust, respect and decision-making discretion support self-determined and agentic individuals, thus promoting thriving [31]. According to the social identity theory, when individuals identify with a group, their self-perception can be formed by the group and the shared attributes that define the group, rather than by their unique individual characteristics [34]. Organizational identity reflects the extent to which an individual has internalized the norms, values and goals of an organization, as well as the quality of embeddedness between them. Employees with high organizational identity are more likely to make challenge appraisals and thrive on job demands.

These findings have some implications for organization management practice. A major implication for practice is that when dealing with job demands such PSDs, employees should be stimulated to focus on the positive sides (i.e., challenge appraisals—seeing it as an opportunity to learn and grow). Leaders should emphasize the potential learning opportunities and benefits brought about by job demands. Cognitive reappraisal training, which is a strategy based on transactional theory whereby employees are taught to understand the impact of their appraisals and re-frame the meaning of a situation as a challenge, can be used in employee training [14]. Second, our results demonstrated that organizational identity can regulate employees' appraisal when experiencing problem-solving demands. Espedido et al. emphasized that the growing interdependence among contemporary employees and the significant influence of environmental factors and team climate on shaping employee attitudes and perceptions should not be disregarded. Their research revealed that psychological safety climate and team problem prevention can moderate the relationship between PSDs and challenge appraisal [6,7]. Consistent with the above studies, we found that when employees have a strong sense of organizational identification, they are more likely to see PSDs as challenges and promote thriving. So, in order to enhance employees' challenge appraisal, organizational leaders can foster an inclusive organizational climate that promotes a learning culture and embraces the uncertainty inherent in problem-solving.

There are some limitations in this study. First, all constructs were measured using self-reported measures which are commonly used in studies. However, this approach may introduce social desirability biases and common method bias. Second, the data were collected at two separate time points with a 4-week interval, with the design making it impossible for us to clarify any causal effects. Third, the Cronbach's $\alpha$ of PSDs and challenge appraisal was low; one possible explanation may be that all participants were from a large state-owned enterprise in China, which has a wide range of businesses. The concept of PSDs were developed in the 1980s because manufacturing technology required employees to solve technical problems such as knife edge wear, overworked parts and programming errors. Therefore, there may be some inapplicability of using PSDs for current occupations. PSDs measurement items can be further modified and refined in the future to reflect contemporary work demands. Future research can consider both organizational factors (e.g., charismatic leadership [35]) and individual factors (e.g., intrinsic motivation [10]; proactive personality [11]; behavioral regulation strategies [8]) as boundary conditions for the cognitive appraisal of PSDs. At the same time, it is considered that the work demands of PSDs relate more directly to problem-solving and innovation [10,11]; individual factors such as goal orientation (i.e., learning, performance-proving and performance-avoidance goal orientation) may have an important impact on appraisal [36].

## 5. Conclusions

Will employees thrive with work demands? Our research has shown that the impact of problem-solving demands on employees thriving at work is contingent upon how these demands are appraised, either as challenges or hindrances. The challenge appraisal of work demands can effectively contribute to employees' thriving at work. The guidance of enterprise managers is crucial in fostering a positive attitude towards job demands among employees. Moreover, we found that organizational identity can enhance employees'



challenge appraisal of problem-solving demands and thereby promote their thriving at work. We recommend that future studies take a more balanced picture by investigating the potential bright sides of work demands on thriving. Because of the limitation of the sample, we encourage researchers to replicate our study with a larger sample of different industries in different countries to improve the generalizability of the results. Moreover, studies are encouraged to investigate the causal effects of work demands on thriving, and other alternative mechanisms and moderators should be tested in the future.

**Author Contributions:** Conceptualization, L.M. and H.M.; methodology, L.M.; software, X.Z.; validation, H.M.; formal analysis, X.Z.; investigation, Y.W.; resources, L.M.; data curation, Y.W. and X.Z.; writing—original draft preparation, L.M.; writing—review and editing, L.M.; visualization, L.M.; supervision, H.M.; project administration, H.M.; funding acquisition, H.M. All authors have read and agreed to the published version of the manuscript.

**Funding:** This research received no external funding.

**Institutional Review Board Statement:** All procedures performed in the study involving human participants were in accordance with the ethical standards of the Central China Normal University, EC, Institutional Review Board (IRB Number: CCNU-IRB-202306003a), as well as with the 1964 Helsinki declaration and its later amendments or comparable ethical standards.

**Informed Consent Statement:** Informed consent was obtained from all subjects involved in the study.

**Data Availability Statement:** Data available within the article. For more information, please contact the corresponding author.

**Conflicts of Interest:** The authors declare no conflict of interest.

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
