# Peer review of "How Do Problem-Solving Demands Influence Employees’ Thriving at Work: An Explanation Based on Cognitive Appraisal"

_sustainability, doi:10.3390/su152014879_

Round 1

Reviewer 1 Report

       The researcher presents an area of concern which is an important emerging topic.

       In depth analysis and a more critical discussion of literature view is recommended. 

       More updated references should be added.

       The social and practical implications/recommendations should be well highlighted and discussed.

Author Response

  1. In depth analysis and a more critical discussion of literature view is recommended. 

Following your suggestion, we provide a more in-depth analysis of the existing research in the introduction. Relevant modifications can be found in the section highlighted in blue.

  1. More updated references should be added.

I'm sorry we haven't been able to update the literature. We conducted an extensive literature search on the most recent relevant studies, and it is evident that there is currently a paucity of research on PSD. We have also attempted to incorporate tangentially relevant literature; however, its inclusion does not significantly bolster our argument and may even introduce unnecessary complexity.

  1. The social and practical implications/recommendations should be well highlighted and discussed.

Following your suggestion, we have explored the practical implications of research in the discussion section. This exploration primarily encompasses two key aspects. Firstly, cognitive appraisal plays important role in influencing employees' diverse responses to stress. Secondly, organizational identity emerges as a crucial factor that fosters employees' challenge appraisal of PSD. Relevant modifications can be found in the section highlighted in blue.

Reviewer 2 Report

Manuscript ID-Sustainability-2602184

In this manuscript (MS) by Lulu Ma et al., studied under title “How Does Problem Solving Demands Influence Employees’ Thriving at Work: an Explanation Based on Cognitive Appraisal”.

This work studied based on the transactional theory of stress, they explored how employees' prospering at work is influenced by how they see their problem-solving responsibilities, whether as a challenge or a burden. Data were obtained from employees of a Chinese state-owned firm at two separate locations over a four-week period. The outcome is satisfactory. This is a fascinating study and advantage. The authors, in my opinion, should elaborate on their findings. The MS may be published after minor revisions. The following comments are suggested to be addressed in the revised MS.

Comments and Suggestions

1.     Table 1: In my opinion, the significance of numbers 1-9 should be specified by the authors at the bottom of the table.

2.     Line 176: It is not necessary to discuss monetary remuneration in this work.

3.     Line 178, 179: What are T1 and T2? What are the distinctions between the T1 and T2?

4.     Line 181: How many men participated in this study? Why do authors just include female details?

5.     Table 1 (Line 226): Some data with high R2 are shown in Table 1, please discuss.

6.     Line 222-225: The author said “Challenge appraisal was significantly positively associated with thriving at work (r=0.598, p<0.1), and hindrance appraisal was significantly negatively associated with thriving at work (r=-0.237, p<0.01). PSD was significantly positively associated with thriving at work (r=0.226, p<0.01)” Please explain why I can't locate these values in table 1. 

7.     Line 293: What is the SEMT?

8.     Discussion (Line 280): The authors mentioned lean production, but there was no explanation elsewhere. It might be preferable to explain briefly. What it has to do with organizational outcomes.

9.     Citation: There is no need to include a page number in the contents. It may appear in the references. Please check line 30, 110.

10.  Citation: The space bar of "[]" should be checked for authors.

Author Response

Thank you for your careful review. Based on your comments, the following modifications have been made.

  1. Table 1: In my opinion, the significance of numbers 1-9 should be specified by the authors at the bottom of the table.

The meaning of the numbers 1-9 was presented in the first column of the table. For the sake of simplicity and aesthetics, the first row of the table no longer displays the meaning of the numbers. We also referenced the table in other literature, and this approach is also acceptable.

  1. Line 176: It is not necessary to discuss monetary remuneration in this work.

As you suggested, we deleted the discussion of specific monetary remuneration. We dded a simple explanation of the subjects' returns as “The participants were appropriately compensated as a token of our gratitude for their valuable contribution.”

  1. Line 178, 179: What are T1 and T2? What are the distinctions between the T1 and T2?

T1 and T2 means Time1 and Time2. Time 1 and time 2 are 4 weeks apart. In order to express clearly and accurately, we have made the following modifications: Demographic variables, PSD, cognitive appraisal and organizational identification were collected in the first questionnaire, and 383 valid data were collected. Four weeks later, we collected thriving in the second questionnaire, and 304 valid data were collected.

  1. Line 181: How many men participated in this study? Why do authors just include female details?

I'm sorry we left out the male ratio. We added the male ratio before female ratio.

  1. Table 1 (Line 226): Some data with high R2 are shown in Table 1, please discuss.

In table 1, there are four high R2 which greater than 0.5.

First, the correlation coefficient between age and employment is 0.55. Because the older you get, the longer you stay in office, it's easy to understand.

Second, the correlation coefficient between challenge appraisal and organizational identification is 0.546. Organizational identity refers that employees “see themselves and the organization as one and empathize with the success and failure of the organization”. Due to the strong link with the organization, employees tend to perceive PSD as an indispensable requirement for personal and organizational development. They are more likely to see the learning and growth brought by problem solving, so they tend to make challenge appraisal. This is consistent with our hypothesis.

Third, the correlation coefficient between challenge appraisal and thriving at work is 0.546. If employees see the opportunities of learning and growth in problem solving, they will strive to learn new skills and constantly try to solve problems. In the process of problem solving, employees continue to learn and maintain vitality to overcome difficulties and solve problems, thus improving thriving at work. This is consistent with our hypothesis.

Fourth, the correlation coefficient between organizational identification and thriving at work is 0.598. When employees’ organizational identity is higher, they have stronger motivation to seek benefits for the organization. In In the face of problem-solving demands, they will remain learning and vitality in order to solve problems and protect the interests of the organization. Therefore, employees with higher organizational identification are more likely to thriving at work.

  1. Line 222-225: The author said “Challenge appraisal was significantly positively associated with thriving at work (r=0.598, p<0.1), and hindrance appraisal was significantly negatively associated with thriving at work (r=-0.237, p<0.01). PSD was significantly positively associated with thriving at work (r=0.226, p<0.01)” Please explain why I can't locate these values in table 1. 

We are sorry that we have lost a row in table 1. The three correlation coefficients you pointed out are marked in blue in the table.

  1. Line 293: What is the SEMT?

Thank you for your comments. SEMT means Socially Embedded Model of Thriving which explains how individuals thrive in environments that enable them to behave agentically at work. We labeled the Socially Embedded Model of Thriving SEMT when it first appeared, which wan marked in blue in the article.

  1. Discussion (Line 280): The authors mentioned lean production, but there was no explanation elsewhere. It might be preferable to explain briefly. What it has to do with organizational outcomes.

As you suggested, we added the relationship between PSD and organizational outcomes in the context of lean production. The following explanation has been added: PSD requires employees to engage in more active cognitive processing to diagnose and solve problems, which serves as a motivator for employees’ job satisfaction [33].

  1. Citation: There is no need to include a page number in the contents. It may appear in the references. Please check line 30, 110.

As you suggested, we deleted the page number.

  1. Citation: The space bar of "[]" should be checked for authors.

As you suggested, we checked the space bar of "[]". We modify the lack or excess of spaces behind or before the dot in the whole paper.

Reviewer 3 Report

Dear authors, 

I read your manuscript and found the study well conducted and properly described. My congratulations for the efforts in scientific research.

Regards,

Author Response

Thank you for your review and your recognition of our research.

Bless you.

Reviewer 4 Report

I am grateful for the possibility to become familiar with a manuscript, which seems to have the potential to become a good publication. I would be happy to review the substance of the analysis and findings. However, before we get to that round, at this stage the following corrections are necessary:

The whole paper must be edited by either a native-English language editor, or maybe at least by some author’s colleagues, or a linguistics/ translation/ philology student. For the time being, it is very difficult, and at times, impossible, to grasp, and hence, to better assess the quality of the substance of the article.

The whole paper must be edited by either a native-English language editor, or maybe at least by some author’s colleagues, or a linguistics/ translation/ philology student. For the time being, it is very difficult, and at times, impossible, to grasp, and hence, to better assess the quality of the substance of the article.

Author Response

Thank you for your review. We asked native English speakers to edit and polish the article. We sincerely invite you to review the article again.

Reviewer 5 Report

The article is interesting. The topic is current, and the literature review is adequate, with a relevant number of contemporary and respectable references. The paper presents a conceptual model which contributes to an overview.  The authors have proven the established hypotheses and correctly explained the methodology. The article explains the study's limitations and recommendations for further research, which is commendable. It is desirable to highlight the limitation of the sample in terms of the number of respondents to make more valid generalizations. This could even be positioned in the conclusions section. Conclusions are not a mandatory part of the paper according to the journal's instructions, but if the authors decided that the conclusions exist as separate ones, they should be enriched with 2-3 sentences, The authors followed the instructions of the journal. The only oversight of the article is the lack or excess of spaces behind or before the dot. It is in the following lines: 40, 54, 65, 74, 77, 84, 86, 87, 91, 93, 96, 102, 112, 118, 154, 279, 288, 293, 296, 299, 327, 328 and 332. In line number 52, there is a semicolon in front of However instead of just a dot. Espedido et al. in the 128th line, it is written whithout spacing. 

Author Response

Thank you for your careful review. Based on your comments, the following modifications have been made.

  1. About “the lack or excess of spaces behind or before the dot”, we unify the format of the references. For example, xxxxx [21,22].That is, a apace before [, but no space between ] and a dot. According to this standard, we have made modifications in the following lines: 40, 54, 65, 74, 77, 84, 86, 87, 91, 93, 96, 102, 112, 118, 154, 279, 288, 293, 296, 299, 327, 328 and 332 which you pointed.

If before [ is also a punctuation mark or another symbol, such as ) or ”, there is no space between the two symbols. According to this standard, we have also made several modifications.

  1. In line number 52, we changed the semicolon to a dot.
  2. Espedido et al. in the 128th line,we add a space.
  3. In accordance with your suggestions, we have supplemented the research conclusions and propose that future researchers replicate and enhance our study using a larger sample encompassing diverse cultures and industries.For details, see the content in blue in the conclusion.

Round 2

Reviewer 4 Report

The paper seems to look better now. However, I am still struggling with the very title of the paper... Should it rather read 

either

How does problem solving demands influence employees’ thriving at work: an explanation based on cognitive appraisal

or

How does problem solving demands influence employees’ thriving at work: an explanation based on cognitive appraisal

?

Yes, please see my above comment on the title.

Author Response

We appreciate your valuable feedback. The title have been modified. How do problem solving demands influence employees’ thriving at work: an explanation based on cognitive appraisal
